# Sensor-Based Assessment of Time-of-Day-Dependent Physiological Responses and Physical Performances during a Walking Football Match in Higher-Weight Men

**DOI:** 10.3390/s24030909

**Published:** 2024-01-30

**Authors:** Sami Hidouri, Tarak Driss, Sémah Tagougui, Noureddine Kammoun, Hamdi Chtourou, Omar Hammouda

**Affiliations:** 1Research Laboratory, Molecular Bases of Human Pathology, LR19ES13, Faculty of Medicine, University of Sfax, Sfax 3026, Tunisia; sami.hidouri@outlook.fr (S.H.); omar.hammouda@parisnanterre.fr (O.H.); 2Interdisciplinary Laboratory in Neurosciences, Physiology and Psychology: Physical Activity, Health and Learning (LINP2), UPL, UFR STAPS, Paris Nanterre University, 92001 Nanterre, France; 3EA7369–URePSSS, Pluridisciplinary Research Unit, “Sport, Health and Society”, University of Lille, University of Artois, University of Littoral Côte d’Opale, 59000 Lille, France; semah.tagougui@univ-lille.fr; 4High Institute of Sport and Physical Education, University of Sfax, Sfax 3000, Tunisia; noureddine.kammoun8@gmail.com (N.K.); hamdi.chtourou@isseps.usf.tn (H.C.)

**Keywords:** recreational soccer, circadian rhythms, HRV, metabolic health

## Abstract

Monitoring key physiological metrics, including heart rate and heart rate variability, has been shown to be of value in exercise science, disease management, and overall health. The purpose of this study was to investigate the diurnal variation of physiological responses and physical performances using digital biomarkers as a precise measurement tool during a walking football match (WFM) in higher-weight men. Nineteen males (mean age: 42.53 ± 12.18 years; BMI: 33.31 ± 4.31 kg·m^−2^) were engaged in a WFM at two different times of the day. Comprehensive evaluations of physiological parameters (e.g., cardiac autonomic function, lactate, glycemia, and oxygen saturation), along with physical performance, were assessed before, during, and after the match. Overall, there was a significant interaction (time of day x WFM) for mean blood pressure (MBP) (*p* = 0.007) and glycemia (*p* = 0.039). Glycemia decreased exclusively in the evening after WFM (*p* = 0.001), while mean blood pressure did not significantly change. Rating of perceived exertion was significantly higher in the evening than in the morning (*p* = 0.04), while the heart rate recovery after 1 min (HRR60s) of the match was lower in the evening than in the morning (*p* = 0.048). Overall, walking football practice seems to be safe, whatever the time of day. Furthermore, HRR60, glycemia, and (MBP) values were lower in the evening compared to the morning, suggesting that evening exercise practice could be safer for individuals with higher weight. The utilization of digital biomarkers for monitoring health status during WFM has been shown to be efficient.

## 1. Introduction

Obesity, often associated with sedentary behavior [1,2], is linked to a high risk of cardiovascular, metabolic, and pulmonary diseases [1,2,3,4,5,6]. This chronic disorder is currently ranked as the 5th highest global public health problem and is responsible for 4.8% of deaths worldwide [7]. Mainly, Tunisian statistics indicate a prevalence of 26.2% for obesity [8]. Particularly, obesity is associated with circadian misalignment, which can be explained by social jet lag and an erratic lifestyle [9]. Indeed, the circadian clock system is involved in controlling the biology of adipose tissue, energy metabolism, and physiological responses to exercise [10]. Due to its chronobiotic effects, exercise can induce a phase shift in the circadian rhythmicity of the supra-chiasmatic nucleus [11,12], which could lessen the harmful effects of circadian misalignment [13,14], and increase the activation of metabolic pathways and systemic energy homeostasis in skeletal muscle [15]. Moreover, exercise is effective in improving hunger suppression as a result of changes in energy balance [16]. For these reasons, regular physical activity is strongly encouraged in order to minimize the consequences of obesity and to fight against other cardiovascular and metabolic disorders [17]. In this context, recreational team sports, particularly football, have emerged as cost-effective interventions promoting cardiometabolic health [18,19,20,21]. Walking football (WF), a variant introduced in 2011, is a feasible and sustainable exercise intervention. WF involves intermittent activity with frequent direction changes and walking pace, following an acyclic pattern. Furthermore, players must always walk and not run during a WF match (WFM). Research has demonstrated that during a 1 h WFM, participants covered a distance of 2.5 miles [22], maintained a walking cadence averaging 44–63 steps per minute [23], and made an average of 100 changes of direction [24]. Research suggests the potential of WF practice to enhance participants’ body composition and physical fitness [25,26], while also contributing positively to mental well-being [27,28,29].

Mainly, it has been demonstrated that timed exercises can have a greater physiological and molecular responses, promoting chronobiological homeostasis and treating conditions like obesity [12,30]. Understanding the time-of-day (TOD)-dependent physiological responses to WFM is crucial for robust evidence-backed guidelines and clinical recommendations.

Regarding cardiovascular responses, previous studies have shown that exercise in the afternoon or in the evening presents a greater glucose reduction than in the morning [31,32]. Moreover, it has been demonstrated that performing 60 min of endurance exercise in the evening resulted in a significant elevation of blood adrenaline levels immediately post-exercise, as compared to the morning [33]. Additionally, previous studies have indicated that cardiovascular responses to exercise are TOD-dependent [34].

Despite substantial evidence of the positive impact of physical activity on reducing mortality risk [34,35,36], knowledge is limited regarding the potential cardiometabolic benefits of WF practice at different TODs.

Recognizing this gap, our focus in this study was on assessing physiological responses and physical performance during a single WFM conducted at different TODs in men with higher body weight. This exploration aims to provide valuable insights into the recommendations on the how and when of exercise, especially in the context of higher-weight individuals.

## 2. Materials and Methods

### 2.1. Participants

Nineteen men volunteered to participate in this study (Figure 1). They were recruited based on the following: (i) they were overweight/obese (BMI > 25 kg/m^2^), (ii) over 30 years old, (iii) they have been sedentary for more than 6 months, and (iv) they did not have any injuries or any cardiovascular, respiratory, or neurological diseases. Answers to the Horne and Ostberg [37] questionnaire categorized subjects as “moderately evening” (*n* = 9), “intermediate” (*n* = 8), or “moderately morning” (*n* = 2) chronotypes.

The characteristics of the present study participants are presented in Table 1. The participants gave written informed consent to participate in the experiment after receiving a detailed description of the possible risks and discomforts related to the experimental procedures. All procedures were approved by the local ethics committee (personal protection committee—CPP SUD, n° 0336/2021, Sfax, Tunisia) and registered in the Pan African Clinical Trial Registry (Trial ID: PACTR202203813725635). This study was conducted according to the Declaration of Helsinki.

### 2.2. Experimental Design

The experimental design consisted of playing 5 vs. 5 WFMs on a field with dimensions ranging from 30 to 45 m in width and 45 to 60 m in length, surfaced with artificial turf, at two different TODs. Participants were familiarized with the WF practice (3 familiarization sessions) before the start of the experimental sessions. Each WF session began with a warm-up of 10 min, and then participants played two 20 min periods interspersed with a 10 min passive recovery. During the recovery period, participants are encouraged to engage in stretching exercises and hydration. The WF session ended with 5 min of stretching (Figure 2). Mini football goals (90 cm × 60 cm) ensure the absence of goalkeepers, fostering an equal playing opportunity for all participants.

Participants have performed two WFMs, in a randomized order, over 2 days with only one test session per day, allowing a recovery period of 7 days in between. The morning session was conducted between 6:30 a.m. and 8:30 a.m., and the evening one was conducted between 5:00 p.m. and 7:00 p.m. Participants were required to arrive 30 min before the commencement of each session. The morning session consisted of pre-match assessments from 06 h 30 to 07 h a.m. and post-match assessments from 08 h to 08 h 30 a.m. The evening session followed the same pattern, with pre-match assessments from 05 h 00 to 05 h 30 p.m. and post-match assessments from 06 h 30 to 07:00 p.m. HR recovery after 1 min (HRR60s) and rating of perceived exertion (RPE) score have been recorded only after the WFM. Moreover, selected physiological parameters and physical performances have been assessed before and after the WFM (Figure 2).

### 2.3. Measurements

Body Composition

A digital scale was used to measure body weight with an accuracy of 0.1 kg (Tanita TBF 401, Tanita Corp., Tokyo, Japan), and a stadiometer was used to measure height with an accuracy of 0.1 cm (Holtain, Crymych, Dyfed, UK). The measured anthropometric parameters were as follows: fat mass (FM), lean body mass (LBM), and total body water (TBW).

Blood Pressure (*BP*)

The systolic and diastolic *BP* were measured with a stethoscope (MissouriR) and a manual aneroid sphygmomanometer (MissouriR) with a precision of 2 mmHg. The measure was taken in a seated position after 10 min of rest, 3 times at intervals of one minute. Mean blood pressure (*MBP*) was calculated with a standard formula as follows [38]:MBP=[diastolic BP+1/3 (systolic BP−diastolic BP)]

The result was taken as the mean value of the three measurements.

Heart Rate Variability (HRV)

We employed a heart rate (HR) sensor (Mooky Center technology, HR5) because of its proven reliability in measuring HR, as demonstrated by Ravier et al. [39]. This device was synchronized with an ActiGraph GT3X (Actigraph, Pensacola, FL, USA) to record the HR and RR intervals. The HR sensor was placed on the chest, and the triaxial accelerometer was placed in the non-dominant wrist, which was calibrated and synchronized to record HR, R-R intervals, and triaxial accelerations. R-R intervals have been recorded using a modified protocol of Kammoun et al. [26], with a sampling rate of 1000 Hz. Then, RR data. csv files were exported from the Actilife 6 (version 6.13.7) app and then processed by Kubios HRV Premium Software (version 3.5). The option “automatic method” was used to correct the RR series [40]. During the analyses, data sets with artifacts >3% were removed from consideration [41]. All HRV measurements are presented in the Table 2.

Oxygen Saturation (SpO^2^)

To measure SpO^2^ levels, a pulse oximeter finger probe (HOL_MD300C15D) was applied to the left hand’s ring finger.

Blood Lactate (La) and Glycemia (Gl) Levels

The levels of blood lactate (La) were assessed using the Lactate Pro 2 electrochemical method portable blood lactate meter (Arkray, Kyoto, Japan), a point-of-care analyzer that functions through enzymatic amperometric detection. The device exhibits reliability, with an intraclass correlation coefficient (ICC) ranging from 0.94 to 0.99 and a coefficient of variation (CV%) between 3 and 3.6% [42]. When a blood sample interacts with the reagent on the test strip, it generates a slight electrical current corresponding to the [La]. The device gauges this slight electrical current and computes the [La]. It only requires 0.3 μL of whole-blood sample and takes 15 sec to measure the lactate value.

Glycemia (Gl) was assessed using the blood glucose test meter (Freestyle, Optium Neo, Avica, Witney, UK Ltd.), which showed a notable ICC of 0.97 [43]. The measurement utilized an electrochemical method, involving a small blood sample taken from the tip of the index finger. Glucose in the blood reacts with an enzyme electrode containing glucose oxidase, which is fixed on the strips, and calculates the glucose level in the blood [44].

Modified Agility T-Test (MAT)

The Modified Agility T-Test (MAT) was used to assess speed with directional changes such as multidirectional sprinting, left and right shuffling, and backpedaling. It has been described in detail elsewhere [45]. The test was timed using a stopwatch (C500B). The MAT has high reliability, with an ICC of more than 0.90 [45].

Vertical Jump Height (VJ)

The explosive strength of the leg muscles was measured using the Takei Vertical Jump Meter (Takei Jump-MD TKK5406, Niigata, Japan), which had high reliability with an ICC of 0.97 and a CV% of 2.58 [46]. Three attempts were made to jump from the standing position without recoil, and the highest score was recorded after this trial was adopted [47,48].

Lumbar Strength (LS)

The measure of the static strength of back muscles was assessed with a Takei back muscle dynamometer (Takei 516489, Tokyo, Japan) with a dial range from 0 to 300 kgs. This shows very high reliability, as evidenced by an ICC ranging from 0.996 to 0.999 [49]. The description of realization was detailed in the study of Kammoun et al. [26]. Three trials were conducted following the demonstration and the familiarization trial, with a 2 min pause in between each trial. Maximal strength for the three trials was used.

Felt Arousal Scale (FAS)

A single item measuring perceived arousal, the FAS [50], ranges from 1 (low arousal) to 6 (high arousal). Arousal is a physiological and mental condition that can be expressed in a variety of ways. High arousal is characterized by excitement, nervousness, or anger, and low arousal is characterized by calmness, relaxation, or boredom [50].

Match Parameters

The ActiGraph GT3X (Actigraph, Pensacola, FL, USA) accelerometer is a reliable and objective measurement tool that has an ICC above 0.925 in terms of vector size [51]. It was used to measure the number of steps (nbS) and the metabolic equivalent of the task (MET) of the subjects during the WFM. The accelerometers’ initial data were collected at a sampling frequency of 30 Hz. The accelerometer was initialized before each test session. Participants wore GT3X activity monitors on their non-dominant arm (Actigraph, Pensacola, FL, USA). Actilife 6 (version 6.13.7) was used to download and analyze the GT3X data. The WFM data of nbS were downloaded to a computer at 60 s intervals, and the predicted expenditure energy was obtained by Freedson’s prediction equation [52].

Heart Rate (HR)

HR data. csv files were exported from Actilife 6 (version 6.13.7), and then Excel software 2019 was used to extract and evaluate HR beats (Microsoft Corporation, Redmond, WA, USA). HR data from the WFM were expressed as mean HR and HR max. We also extracted the mean HR after 1 min from the WFM (HRR60s).

Rating of Perceived Exertion (RPE)

Participants rated their subjective level of exertion for physical exercise using the Borg CR10 RPE scale, which has values ranging from 0 (no effort) to 10 (maximal effort) [53].

### 2.4. Sample Size Calculation

The study was designed as a counter-balanced trial, where each participant was exposed to both experimental conditions (morning exercise compared to evening exercise). Furthermore, the minimum required sample size was calculated using the G*power software (version 3.1.9.6; Kiel University, Kiel, Germany). We set α and power (1-β) values at 0.5 and 0.9, respectively. The sample size of the study was calculated, a priori, based on the study of Brito et al. [54]. To achieve the desired power, data from 16 participants would be sufficient to minimize the risk of incurring a type 2 statistical error. Twenty volunteers have been included to accommodate any potential dropouts during the experiments. Finally, we completed the experimentation with 19 participants, as one individual sustained an injury and did not complete the study.

### 2.5. Statistical Analysis

Values are reported as the mean ± standard deviation (SD). SPSS version 25 for Windows (SPSS Inc., Chicago, IL, USA) was used to perform data analyses. The Shapiro–Wilk test was used to determine if the data were normally distributed. A two-way analysis of variance was used to determine differences between experimental conditions (2 times of day (morning and evening) × 2 time points: before and after the WFM). The Bonferroni post hoc test was applied when a significant difference was found. However, when normality failed, a Wilcoxon test was used to compare all conditions and to determine the interaction TOD × delta change match (after–before match). To estimate the meaningfulness of significant findings, the effect sizes were calculated as partial eta-squared (ηp2) for the ANOVA. Small, moderate, and large effect sizes were represented by ηp2 values of 0.01, 0.06, and 0.13, respectively. For nbS, mean HR, MET, HR max, HRR60s, and RPE, we used the paired student *t*-test when data were normally distributed to analyze the effect of TOD (morning versus evening). The Wilcoxon test was used for non-parametric data. Cohen’s d test was utilized to determine effect sizes, with values typically interpreted as follows: a small effect (0.2), a medium effect (0.5), and a large effect (0.8) or higher [55]. Statistical significance was considered when *p* < 0.05.

## 3. Results

### 3.1. HRV

Table 3 displays the statistical results of the diurnal variation of HRV data (time domain and frequency domain parameters) before and after WFM.

Time domain

Statistical analyses revealed no significant interaction (TOD × match) for HR and mean RR, RMSSD, and SDNN.

Frequency domain

For the frequency parameters, there was no significant interaction (TOD × match) for LF, HF, and the HF/LF ratio.

### 3.2. Physiological Parameters

For the physiological parameters (Table 4 and Figure 3), statistical analyses showed a significant interaction (TOD × match) only for (Gl) (*p* = 0.039; ES = 0.21) and MBP (*p* = 0.007; ES = 0.34). The Wilcoxon test revealed that (Gl) decreased only in the evening after WFM (*p* = 0.001), while MBP did not show any significant change. There was no significant interaction (TOD × match) for (La) and SpO^2^.

### 3.3. Physical Parameters

As shown in Table 4 and Figure 4, no significant interaction (TOD × Match) was observed for physical parameters, especially VJ, MAT, and LS.

### 3.4. Psychological State

Statistics analyses showed no significant interaction (TOD × Match) for FAS (Table 5).

### 3.5. Match Parameters

Statistics analyses showed that no significant difference was observed between the two TODs (Table 6 and Figure 3 and Figure 4) in nbS (*p* = 0.54, d = 0.17), mean HR (*p* = 0.40, d = 0.19), MET (*p* = 0.18, d = 0.08), and HR max (*p* = 0.60, d = 0.03). But HRR60s from the end of the match and RPE presented a remarkable difference between the evening and morning. The HR decreased quickly in the evening compared to the morning (*p* = 0.048, d = 0.47). Also, RPE was lower in the morning compared to the evening (*p* = 0.04; d = 0.47).

## 4. Discussion

To the best of the authors’ knowledge, this is the first study to investigate the diurnal fluctuation of physiological responses and physical performance using biosensors before and after practicing WFM in higher-weight individuals.

Despite the numerous beneficial effects of regular physical activity practice [56], acute strenuous exercise was associated with an increased risk of cardiovascular events [1,4]. Therefore, it is crucial to thoroughly investigate these exercise-related factors to maximize the benefits of exercise and prevent the occurrence of ischemic events, especially in obese and overweight individuals. Monitoring cardiometabolic parameters using sophisticated sensors that offer digital health biomarkers (e.g., HR, HRV, HRR60s, glycemia, and SpO_2_) is essential for maintaining overall health.

In this context, HRV is a useful tool in assessing the risk of cardiac events and in evaluating the advantages and disadvantages linked to physical exercise [57]. Indeed, persons living with obesity showed a decrease in HRV, indicating an alteration in cardiac autonomic function [58], due to the predominance of sympathetic tone [59]. Moreover, the importance of HR assessment in the recovery phase as a predictor of mortality has been taken into consideration [60].

According to previous recommendations, individuals should aim for 100 steps per minute while walking to achieve a reasonable cadence for moderate-intensity exercise [61]. Current wearable devices can measure and quantify the number of steps that indicate the varying levels of daily activity. Therefore, it was crucial to examine the nbS walked during a WFM to gain insights into the intensity of this activity. For this reason, we used the accelerometer (Actigraph, GTX3) to measure the nbS and to determine the MET during the WFM.

### 4.1. Physiological Parameters

Our investigation revealed a lack of significant interaction between TOD and the WFM for time domain parameters, including mean RR, SDNN, and RMSSD. This finding aligns with a prior study that found no significant differences in RMSSD before and after a 30 min recovery period in the morning, afternoon, and night in sedentary healthy men after 35 min of cycling [62]. However, another study conducted on men with pre-hypertension displayed a slower RMSSD30s recovery after maximal exercise performed in the evening compared to the morning [54]. These variations underscore the intricate interaction between TOD, exercise, and autonomic nervous system responses, suggesting that the impact may be context-dependent and influenced by individual health status.

Same, frequency domain parameters did not show any significant interaction TOD × match. Accordingly, previous research concluded that cardiac autonomic control remained constant across the different TODs, as indicated by HRV analysis, during the 24 h period following moderate aerobic exercise in sedentary subjects [62,63]. HRV can be affected by many factors, including physical activity, independently of TOD.

In the present study, HRV biomarkers were influenced by the WFM. Indeed, after ceasing exercise, both HR and HRV exhibit a time-dependent recovery, ultimately reverting to their pre-exercise levels [64]. Limited studies have examined how exercise intensity and duration affect HRV during the first 10 min of recovery after a workout. In this sense, Michael et al. [65] suggest that exercise intensity might have a subtle effect on HRV recovery, especially during the first hour after exercise. Notably, another study found that higher exercise intensity resulted in slower recovery of HR and HRV [65]. In conclusion, TOD did not affect HR and HRV recovery after WFM.

In this current investigation, the significant interaction TOD × match was observed for MBP. The WFM-induced increase in MBP was more pronounced in the morning compared to the evening. Accordingly, a prior study has reported elevated MBP in the morning compared to the evening after exercise in normotensive males [66]. In contrast, another study in healthy subjects showed no significant changes in BP after walking and running exercises [67]. Moreover, previous research did not find significant differences in HR and BP values between morning and evening after a 30 min cycling session [66]. According to Whelton et al. [68], BP variability is regulated by the sympathetic nervous system. Additionally, earlier studies have indicated that forearm blood flow after exercise reaches its highest in the afternoon (from 3 p.m. to 7 p.m.) [69]. Moreover, compared to morning exercise (at 8 a.m.), another study found a progressive increase in systemic vascular resistance after afternoon exercise (at 4 p.m.) [70]. These observations explain the faster decrease in BP in the afternoon, which supports our findings.

Although we did not find any significant interaction TOD × match for HRV parameters, the present findings showed that HRR60s was lower in the evening than in the morning. These results can be explained by the findings of Molina et al. [71], which indicated no significant correlation between HRR and HRV in the supine position during the first minute after exercise. Lower HRR60 values in the evening can be explained by the function of sympathetic activity, which rose in the morning and reached its lowest during the evening [72]. Previous studies have also indicated that cardiovascular responses to exercises are TOD-dependent [34]. According to previous research, the most marked peaks in catecholamine reactivity were around 9 h 00 a.m. and 9 h 30 p.m., a decline in vagal activity around 9 h 00 a.m., and a faster recovery of systolic BP after exercise observed in the late afternoon (around 5 h 00 p.m.) compared to the early morning (8 h 30) [73]. Some studies have established a threshold of 12 beats per minute after exercise to define a low HRR value [74], which is the difference between peak exercise HR and HRR60 [75]. In this context, in a previous study, HRR60s were found to be similar after maximal exercise in the morning (27 ± 7 bpm) and evening (29 ± 7 bpm) among hypertensive men [54]. Notably, a correlation exists between mortality risk and low HRR values [74]. All these findings contribute to the understanding of the safety of exercise based on HRR values.

Otherwise, means of (La) levels increased after the WFM at both morning and evening sessions. This finding is consistent with a previous study that evaluated (La) levels before and after WFM in older individuals, revealing a significant increase of around 157% post-session [24]. This highlights the high intensity of WFM, as evidenced by the comparatively high post-session (La) concentration. These findings suggest that WF practice places a significant demand on the glycolytic pathway in addition to the aerobic ones.

Our results showed a significant interaction between TOD × match for (Gl) levels. The decrease in (Gl) levels after the WFM was higher in the evening than in the morning. This aligns with previous studies that reported reduced (Gl) levels after evening exercise of similar intensity [76,77]. Furthermore, catecholamines, the main hormones whose concentrations increase markedly during exercise [33], play a pivotal role in elevating lipolysis and glycogenolysis [78]. Additionally, during exercise, muscle fibers produced IL-6 [79], enhancing both (Gl) uptake and fat oxidation [80,81]. Notably, it has been indicated that catecholamines and IL-6 exhibit fluctuations throughout the day, with higher levels observed in the evening compared to the morning [78]. The above-mentioned mechanisms could support greater blood (Gl) control and insulin sensitivity after evening exercise more than morning exercise [31,82]. Moreover, the participant’s psychological condition did not vary significantly between morning and evening sessions, as assessed by the FAS. From the FAS results, it seems that the diurnal variation of some physiological parameters in the present study was not dependent on the psychological state of the participants.

### 4.2. Physical Parameters

There was no significant interaction TOD × match for the physical parameters measured in the present study. MAT did not reveal any notable differences. However, a previous study noted an average of 100 changes in direction and 45 accelerations and decelerations at a speed of 1.5 m/s² during a WFM [24]. A systematic review suggests that anaerobic power (i.e., jump height) peaks in the afternoon [83]. However, in our current study, we did not observe a significant interaction between TOD × match. It appears that the duration of the WFM has suppressed the impact of TOD on physical performance (MAT, LS, and VJ).

The present study is the first to consider the nbS taken during a WFM among higher-weight men. Our statistical results did not reveal any TOD effect on the nbS. During the WFM, participants achieved an average of 2546.94 ± 412.37 steps in the morning and 2611.38 ± 356.34 steps in the evening. These findings align with another study, which indicated that obese men normally walk at a cadence of around 63.8 steps/min [84]. Prior suggestions state that walking at a pace of 100 steps per minute is ideal for stimulating moderate intensity [61]. Therefore, participating in WFM can assist high-weight men in meeting the daily step count recommendations and enable them to achieve the required amount of moderate-to-vigorous intensity activity.

### 4.3. Match Parameters

In our study, we did not find any difference in MET values (4.85 ± 0.62 in the morning and 4.9 ± 0.46 in the evening) during WFM, which corresponds to the moderate exercise intensity category according to the established MET classification, where activities are classified as light intensity (<3 METs), moderate (3–6 METs), or vigorous (>6 METs) [85,86]. Additionally, we did not find any significant change in mean HR at both TODs (133.54 beat/min in the morning and 136.54 in the evening). The fact that the morning and evening values are similar suggests that there has not been any change in the level of physical activity throughout the day. indicates that the variations in blood glucose and HRR60 are not related to the variability of exercise intensity between both sessions. Moreover, HR max recorded during the WFM did not change across TOD. A previous study no significant difference between morning and evening in mean HR and HR max in healthy subjects during progressive cycling exercise [87].

Higher HR max and mean HR values are in line with findings of previous studies on recreational small-sided football training [19] and WF sessions [24] in elderly men. Furthermore, according to Harper et al. [24], WF sessions exhibit a relative intensity of 90% VO_2peak_ in the elderly. Indeed, WF has been classified as an intermittently vigorous-intensity exercise for the elderly [28]. The present findings support the claim that football-based training at these intensities has major positive effects on the cardiovascular system [19].

### 4.4. RPE

The present study revealed that RPE values were significantly lower in the morning compared to the evening. The RPE values obtained during the WFM indicated a moderate (≥4) to very hard intensity (≥7) level. These RPE values are similar to those reported in a prior study conducted on WF in older adults [28]. To improve and maintain cardiorespiratory and musculoskeletal capacity, the American College of Sports Medicine suggests practicing moderate-intensity exercises corresponding to RPE = 13 (equivalent of %HR max = 64–76% and %VO_2_max 46–63%) [88]. Based on our findings, WF can be considered a safe activity that has the potential to improve the cardiorespiratory and metabolic health of individuals living with obesity.

### 4.5. Limits

The present study presents some limits: Firstly, the participants’ pool was limited in number. Secondly, the demographic characteristics are not well detailed (there are huge differences between the patients). Lastly, a single session is not enough; more sessions are needed to confirm the results.

## 5. Conclusions

The application of digital biomarkers for monitoring health status during WFM has been shown to be an effective and insightful approach. Notably, our study revealed a lack of significant diurnal variation in physiological parameters (HRV indices, lactate, and SpO_2_) and physical performance.

In contrast, some specific metrics, such as HRR60s and MBP, were significantly lower in the evening compared with the morning. The reduction in (Gl) levels was greater in the evening compared to the morning. Overall, the safety of WF practice for higher-weight men has been demonstrated. Furthermore, it appears that practicing WF in the evening is safer, as indicated by HRR60, (Gl), and MBP findings.

As we move forward, future studies should focus on investigating the effects of long-term WF practice, particularly concerning cardiovascular health and metabolism. A more extensive exploration of these aspects will contribute valuable insights for optimizing the health benefits of walking football, especially for individuals with a higher body weight.

## Figures and Tables

**Figure 1 sensors-24-00909-f001:**
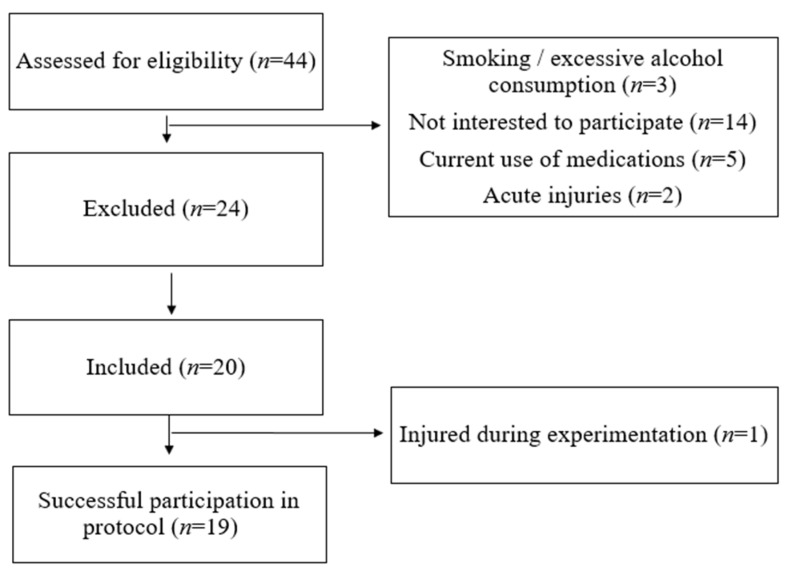
Flow chart of participants’ recruitment.

**Figure 2 sensors-24-00909-f002:**
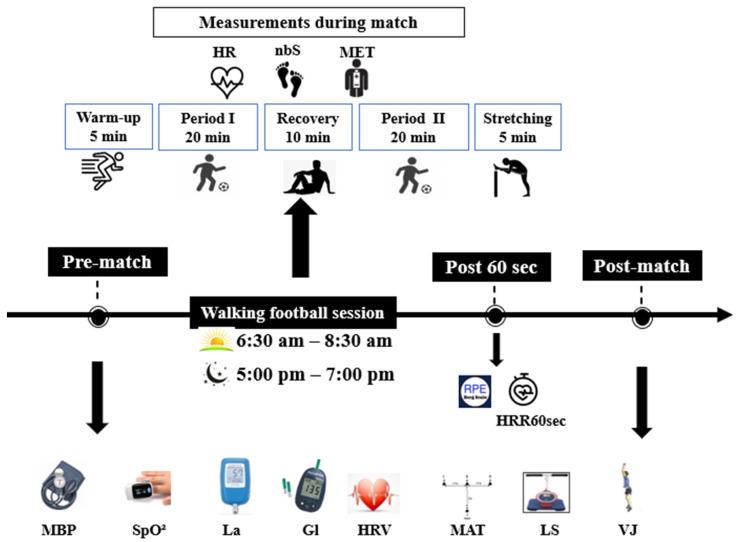
Experimental protocol. HRV: heart rate variability; MBP: mean blood pressure; SpO^2^: oxygen saturation; La: blood lactate; Gl: glycemia; MAT: modified agility T-Test; VJ: vertical jump height; LS: lumbar strength; nbS: number of steps; HR: heart rate; MET: metabolic equivalent of task; HRR60s: heart rate of recovery after 60 s; RPE: rating of perceived exertion; and ↓: measurement.

**Figure 3 sensors-24-00909-f003:**
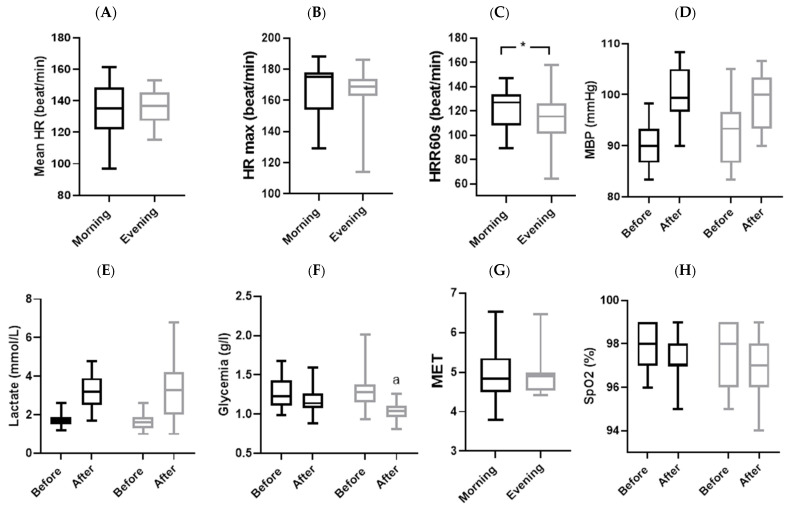
Group means and standard deviation of physiological parameters. (**A**): mean HR: mean heart rate; (**B**): HR max: maximal heart rate; (**C**): HRR60s: heart rate of recovery after 60 s; (**D**): MBP: mean blood pressure; (**E**): lactate; (**F**): glycemia; (**G**): MET: metabolic equivalent of task; (**H**): SpO^2^: oxygen saturation; *: significantly different at *p* < 0.05; a: significantly different from before at *p* < 0.05; black plot: morning and gray plot: evening.

**Figure 4 sensors-24-00909-f004:**
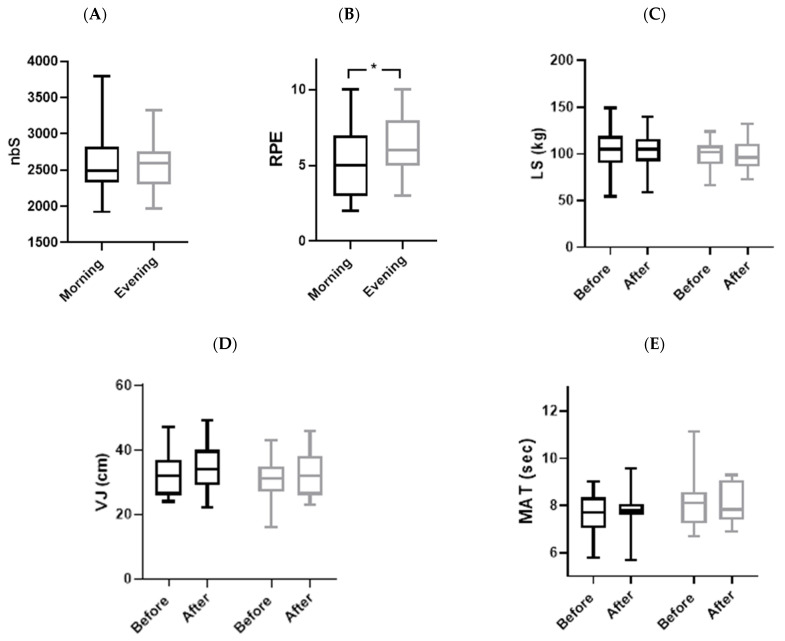
Group means and standard deviation of physical parameters. (**A**): nbS: number of steps; (**B**): RPE: rating of perceived exertion; (**C**): LS: lumber strength; (**D**): VJ: vertical jump; (**E**): MAT: modified agility test; a: significantly different from before; *: significantly different at *p* < 0.05; black plot: morning and gray plot: evening.

**Table 1 sensors-24-00909-t001:** Participants anthropometric characteristics.

Sex	Male
Age	44.89 ± 6.51
Height (cm)	173.16 ± 4.31
Weight (kg)	100.16 ± 13.47
BMI (kg/m^2^)	33.16 ± 4.75
Fat mass (kg)	33.79 ± 10.19
Lean mass (kg)	66.37 ± 4.67
Body water (kg)	47.55 ± 6.99

**Table 2 sensors-24-00909-t002:** HRV measurements.

Parameters	Unit	Description	Indication
Time domain measurements			
**RMSSD**	milliseconds: ms	Root mean square of the successive differences of the R-R intervals	Contribution of variations at high frequencies, which are in turn associated with vagal activity
**SDNN**	milliseconds: ms	Standard deviation of NN intervals	Participation of all rhythmic components responsible for variability, being related to contributions from both branches of the autonomic nervous system
**Frequency domain measurements**			
**LF**	normalized units: n.u	Relative power of the low-frequency band (0.04–0.15 Hertz) in normal units	Baroreceptors activity during resting conditions and parasympathetic and sympathetic nervous systems activity
**HF**	normalized units: n.u	Relative power of the high-frequency band (0.15–0.4 Hertz) in normal units.	Parasympathetic activity
**LF/HF ratio**	—	Ratio of LF to HF power	Sympathetic–parasympathetic balance

**Table 3 sensors-24-00909-t003:** Variation of HRV averages (time domain and frequency domain parameters) before and after WFM in two moments of the day (morning vs. evening) (values are means ± SD).

		Means ± SD			Interaction Time of Day × Match		
		Before	After	Δ	F/Z	*p*-Value	ηp2/ Cohen’s d
**Mean HR (beat/min)**	Morning	74.4 ± 5.28	86.8 ± 8.71	12.39 ± 10.29	1.825	0.194	0.092
	Evening	76.42 ± 5.78	93.27 ± 10.17	16.85 ± 12.09			
**Mean RR (ms)**	Morning	810.1 ± 54.8	698 ± 71.5	−112.1 ± 89.4	0.847	0.370	0.045
	Evening	789.4 ± 59.4	651 ± 75.3	−138.4 ± 97.4			
**SDNN (ms) ***	Morning	38.79 ± 23.43	30.63 ± 11.64	−8.16 ± 27.89	0.644	0.520	0.148
	Evening	37.51 ± 19.68	23.68 ± 10.02	−13.83 ± 17.01			
**RMSSD (ms) ***	Morning	32.01 ± 22.03	19.68 ± 7.95	−12.33 ± 24.87	1.248	0.212	0.286
	Evening	32.14 ± 23.6	16.89 ± 10.7	−15.25 ± 22.35			
**LF (n.u.) ***	Morning	59.97 ± 20.88	72.71 ± 15.71	12.73 ± 22.79	1.327	0.184	0.304
	Evening	65.27 ± 17.45	68.9 ± 17.29	3.63 ± 20.11			
**HF (n.u.) ***	Morning	39.97 ± 20.86	27.24 ± 15.66	−12.73 ± 22.76	1.248	0.212	0.286
	Evening	34.67 ± 17.41	31.06 ± 17.27	−3.61 ± 20.03			
**LF/HF_ratio ***	Morning	2.46 ± 2.2	4.09 ± 3.08	1.63 ± 4.03	0.885	0.375	0.203
	Evening	3.03 ± 2.8	3.38 ± 2.5	0.36 ± 3.34			

Δ: difference between after and before; *: non-parametric test (Wilcoxon test, with Cohen’s d as an effect size).

**Table 4 sensors-24-00909-t004:** Diurnal variation of physiological and physical parameters, before and after WFM (values are means ± SD).

		Means ± SD	Interaction TOD × Match		
Parameters	TOD	Δ	F/Z	*p*-Value	ηp2/ Cohen’s d
**MBP (mmHg)**	Morning	9.88 ± 5.33	9.275	0.007	0.340
	Evening	5.61 ± 6.94			
**SpO^2^ (%) ***	Morning	−0.74 ± 1.24	0.659	0.510	0.151
	Evening	−1.11 ± 1.91			
**Lactate (mmol/L)**	Morning	1.48 ± 0.95	0.310	0.585	0.017
	Evening	1.63 ± 1.13			
**Glycemia (g/L)**	Morning	−0.08 ± 0.2	4.960	0.039	0.216
	Evening	−0.27 ± 0.29			
**MAT (s) ***	Morning	0.08 ± 0.48	1.087	0.277	0.249
	Evening	−0.17 ± 0.84			
**LS (Kg)**	Morning	−0.71 ± 21.87	0.003	0.955	0.000
	Evening	−0.71 ± 13.08			
**VJ (cm)**	Morning	2 ± 3.42	0.018	0.894	0.001
	Evening	1.84 ± 4.52			

Δ: difference between after and before; VJ: vertical jump; MAT: modified agility test; LS: lumber strength; SpO_2_: oxygen saturation; MBP: mean blood pressure; lactate; glycemia; and *: non-parametric test (Wilcoxon test, with Cohen’s d as an effect size).

**Table 5 sensors-24-00909-t005:** Variation of psychological state before and after WFM in two moments of the day (morning vs. evening) (values are means ± SD).

		Means ± SD			Interaction Time of Day × Match		
		Before	After	Δ	Z	*p*-Value	Cohen’s d
**FAS**	Morning	3.32 ± 0.95	2.89 ± 0.74	0.42 ± 0.84	0.104	0.91	0.00
	Evening	3.37 ± 0.96	2.95 ± 1.22	0.42 ± 1.02			

FAS: felt arousal scale; Δ: difference between after and before.

**Table 6 sensors-24-00909-t006:** Match parameter results (mean ± SD).

Parameters	t/Z	*p*	Confidence Interval −95%	Confidence Interval 95%	Cohen’s d
**RPE**	−2.22	0.04	−2.05	−0.06	0.47
**nbS**	−0.63	0.54	−281.95	153.06	0.17
**MET ***	1.33	0.18	-	-	0.08
**Mean HR (beat/min)**	−0.86	0.40	−9.43	3.96	0.19
**HR max * (beat/min)**	0.52	0.60	-	-	0.03
**HRR60s * (beat/min)**	2.29	0.048	0.073	16.091	0.47

nbS: number of steps; mean HR: mean heart rate; MET: metabolic equivalent of task; HR max: maximal heart rate; HRR60s: heart rate of recovery after 60 s; and RPE: rating of perceived exertion in the morning and in the evening. *: non-parametric test.

## Data Availability

The data sets examined in this study can be obtained from the corresponding author upon a reasonable request.

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
