# Peer review of "Sensor-Based Assessment of Time-of-Day-Dependent Physiological Responses and Physical Performances during a Walking Football Match in Higher-Weight Men"

_sensors, 2024, doi:10.3390/s24030909_

Round 1

Reviewer 1 Report

Comments and Suggestions for Authors

Sensor-based assessment of time-of-day dependent physiological responses and physical performance during a walking football match in higher-weight men

General comments to the authors

First, the reviewer would like to thank the authors for their work and efforts in improving sports science knowledge.

Overall, the article is an interesting approach to investigating the diurnal variation of physiological responses and physical performances using digital biomarkers as a precise measurement tool during a walking football match (WFM) in higher-weight men. The study is well-designed and well-written, with a great introduction proposing the usefulness of the topic and a clear outline of the research question. This study also could have potential practical applications, especially in popular team sports like soccer. However, I suggest that the author modify/include some suggestions to improve the manuscript before being potentially published:

I suggest only small corrections:

How many years of sedentary lifestyle do they have? Please add it to the methods

ICC for tests should be added

A number of ethical files should be added.

What kind of surface? It should be added.

Goalkeeper excluded? Please add this information

Is there any G-power analysis? Please add it. If your sample size is small, please add it to the limitations.

Effect size descriptors (small, medium or large) should be added

Please add a limitation section

Overall, the manuscript is well-designed and well-written and incorporates relevant literature.

Author Response

Authors' point-by-point responses to the reviewers’ comments

Manuscript Number 2758009

The authors would like to thank the reviewers for the insightful and constructive comments on our work. We have carefully considered all of the suggestions and have revised the manuscript accordingly. We believe that the revised version of our manuscript is clearer now.

Please find the authors’ responses to the individual comments below. We have highlighted the changes within the manuscript.

Response to Reviewer 1:

  • How many years of sedentary lifestyle do they have? Please add it to the methods

Response: Thanks for this remark. This information has been added to the inclusion criteria in the section 2.1 participants: “they have been sedentary for more than 6 months”.

  • ICC for tests should be added

Response:

Thank you for this comment. We have carefully considered it in the revised version. ICCs of tests have been added as follow:

  • Lactate Pro2: “The device exhibit’s reliability with an ICC ranging from 0.94 to 0.99 and a CV% be-tween 3 and 3.6%.” Please see reference [1] in the bottom of this text.
  • Glycemia: “…which showed a notable ICC of 0.97.” Please see reference [2] in the bottom of this text.
  • Modified Agility T-Test: “The MAT has high reliability, with an ICC higher than 0.90.”Please see reference [3] in the bottom of this text.
  • Vertical jump height: “which had a high reliability with an ICC of 0.97 and a CV% of 2.58. Please see reference [4] in the bottom of this text.
  • Lumbar Strength: “This shows a very high reliability, as evidenced by an ICC from 0.996 to 0.999.” Please see reference [5] in the bottom of this text.
  • The ActiGraph GT3X: “is a reliable and objective measurement tool that has an ICC above 0.925 in terms of vector size. Please see reference [6] in the bottom of this text.
  • We have used a HR sensor (Mooky center technology, HR5) based on the study of Ravier et al. (2022) which showed that there was no difference in HR measured with the device in two separate sessions of small-sided games. This finding demonstrated the reliability of this device. Please see reference [7] in the bottom of this text. 

We added this sentence: “We employed a HR sensor (Mooky center technology, HR5) because of its proven reliability in measuring heart rate, as demonstrated by Ravier et al. (2022)”. Please see reference [7] in the bottom of this text. 

  • A number of ethical files should be added.

Response:  The number of ethical files has been added in line 95: “n° 0336/2021, Sfax, Tunisia”

  • What kind of surface? It should be added.

Response: The kind of surface has been added in the section 2.2. Experimental design : “…, surfaced with artificial turf”

  • Goalkeeper excluded? Please add this information

Response: A sentence has been added in the section 2.2. Experimental design: “Mini-football goals (90cm x 60cm) ensure the absence of goalkeepers, fostering an equal playing opportunity for all participants”

  • Is there any G-power analysis? Please add it. If your sample size is small, please add it to the limitations.

Response: Thank you for this comment. In the original version, we have detailed the calculation of sample size in section 2.4 Sample Size Calculation. The number of participants was mentioned as a limit in the section 4.5. Limits (line 425).

  • Effect size descriptors (small, medium or large) should be added

Response: Thank you for this comment. In the present study we have two types of effect size, the first calculated as partial eta-squared (ηp2) from the Anova results is detailed in line 240. The second was calculated as Cohen's d form post hoc analysis and is mentioned in line 245 in the section 2.5. Statistical Analysis. Descriptors have been detailed in the same section.

  • Please add a limitation section

Response: A title has been added to the limits as 4.5. Limits

Reviewer 2 Report

Comments and Suggestions for Authors

The manuscript presents a statistical analysis to assess the relationship between physiological and physical measurements and time-of-day football practice in higher-weight men. In general, the reviewer agrees with the methodology and the manuscript’s outline. However, the following aspects require major revision:

·       The reviewer is not confident that “the relation between recreational exercise (e,i, WF) at different TOD and health risks” has still not been covered in the literature. Detail on related studies and open-research challenges according to the manuscript’s goal.

·       The participants performed 2 match days. However, it may not be enough to find statistically significant results about time-of-the day. Do the intra-subject results were consistent over 2 days? The participant’s psychological condition may be different over 2 days, which may affect the results’ significance. Discuss this aspect in the manuscript.

·       Variations in physiological and physical measurements may depend on the intensity of the activity (usually, refereed in terms of traveled distance, number of sprints, and number of steps). Are the inter-subject results affected by the intensity of the activity? How do you ensure that variations in glycemia, perceived exertion, and HRR60s are not related to the higher/lower intensity of the activity? Detail this aspect in the manuscript.

1.     Abstract: The abstract includes several abbreviations, which are difficult to read and consequently to interpret the results. Please, reduce the use of abbreviations throughout the manuscript.

2.     Introduction:

2.1 Line 36: It states two times the term “associated”. Please, rephrase the sentence.

2.2 Line 52: Indicate the meaning and/or the differences of walking football when compared to regular walking. Refer to the speed and cadence differences.

2.3 Lines 68-70 state the gap in the literature that this paper aims to address. However, the studies 32 and 33 are not from the last years. The reviewer is not confident that “the relation between recreational exercise (e,i, WF) at different TOD and health risks” has still not been covered in the literature. Detail on related studies and open-research challenges according to the manuscript’s goal. Line 70: Attention to the term “e,i,”. The authors are encouraged to perform a careful English revision.

3.     Methods:

3.1  Provide a rationale for including only participants aged 30 years old. Complete the inclusion criteria regarding the gender, since the participants only include male subjects. Complete Table 1 with information about the exclusion criteria, i.e., indicate the number of participants excluded for each criterion.

3.2   Line 86: replace “de-tailed”.

3.3  Detail the procedures of “10-minute passive recovery”. Were the participants free to self-select the activities during the resting period?

3.4  Provide a rationale for the selected schedule of the times of day. Indicate the assessment time for pre-match and post-match.

3.5  Measurements: indicate the sampling frequency and if they were synchronized. Indicate the sensor system used to measure the “modified agility t-test”. What is nbs in line 173? How was energy expenditure computed from accelerometer data? The authors should revise section 2.3 to detail the meaning of all measurements, the equipment, and the procedure used for their acquisition.

3.6  Indicate the level of significance.

3.7  In line 190, the authors refer to a controlled parallel trial. However, based on the text reported in section 2.2., it seems that all participants performed the four matches (2 time-of-day periods for 2 days). Clarify this aspect in the manuscript.

4.     Results

4.1  Revise the section to avoid misreading. In Line 220, verify the number of table. In line 224, the term “a significant” is reported without reference to a measure. In Table 3, the footnote symbols are not indicated within the results (lines).

4.2  The reviewer recommends the authors to highlight the significant values in Tables 3 and 4. It would facilitate the results’ interpretation.  

5.     Discussion

5.1  Use of the term “Interaction” instead of “relationship”  

5.2  Line 352: replace “the present” by “The present”.

5.3  Remove the term “However” in line 390. It does not seem related to the previous sentence.

The authors stated that “While the heart rate recovery after 1 minute (HRR60s) of the match was lower in 28 the evening from the morning (p=0.048). In conclusion, the present study demonstrated the safety 29 of practicing WFM at different TOD.” Please, detail how reached such a conclusion from the achieved results.  

Comments on the Quality of English Language

Please, revise the manuscript. Some aspects were mentioned in the previous comment section. Commas are missing in various sentences.  

Author Response

Authors' point-by-point responses to the reviewers’ comments

Manuscript Number 2758009

The authors would like to thank the reviewers for the insightful and constructive comments on our work. We have carefully considered all of the suggestions and have revised the manuscript accordingly. We believe that the revised version of our manuscript is clearer now.

Please find the authors’ responses to the individual comments below. We have highlighted the changes within the manuscript.

Response to Reviewer 2:

  • The reviewer is not confident that “the relation between recreational exercise (e,i, WF) at different TOD and health risks” has still not been covered in the literature. Detail on related studies and open-research challenges according to the manuscript’s goal.

Response: Thank you very much for your comments. We are aware that some previous studies have investigated the effects of physical activity at different times of the day, but they have reported conflicting results (Weitzer et al. 2021 [8], Shen et al. 2023 [9], Janssen et al. 2022 [10] and Feng et al. 2023 [11]). The discrepancy between studies could be explained by the difference in the studies’ participants and experimental designs. However, our study focused on a specific type of exercise (walking football) and a specific population (men with high body weight), which have not been well studied in this context. Based on a rigorous literature review, we did not find studies that have been interested in assessing the acute response to walking football with respect of diurnal variation. Please see below the search strategy we used for literature.

PubMed

[1]

“walking football” [tw] OR “walking soccer” [tw] OR “recreational football” [tw] OR “recreational soccer” [tw]

[1] AND [2]

N= 0

N= 189

[2]

"time of day"[tw] OR "chronotype"[tw] OR "chronobiology"[tw] OR "diurnal variation"[tw]

N= 19,086

Google scholar

[1]

“walking football” [tw] OR “walking soccer” [tw] OR “recreational football” [tw] OR “recreational soccer” [tw]

[1] AND [2]

N= 0

N= 7

[2]

"time of day"[tw] OR "chronotype"[tw] OR "chronobiology"[tw] OR "diurnal variation"[tw]

N= 481

Wiley

[1]

“walking football” OR “walking soccer” OR “recreational football” OR “recreational soccer”

[1] AND [2]

N= 7

(N=0 After filtration)

N= 159results

[2]

"time of day" OR "chronotype" OR "chronobiology" OR "diurnal variation"

N= 67 339

Scielo

[1]

walking football OR walking soccer OR recreational football OR recreational soccer

[1] AND [2]

N= 0

N= 22

[2]

time of day OR chronotype OR chronobiology OR diurnal variation

N= 2 342

  • The participants performed 2 match days. However, it may not be enough to find statistically significant results about time-of-the day. Do the intra-subject results were consistent over 2 days? The participant’s psychological condition may be different over 2 days, which may affect the results’ significance. Discuss this aspect in the manuscript.

Response:  Thank you for your comments. We appreciate your valuable insights and observations. In the present study, we focused on exploring the influence of diurnal variations on our study variables. To prevent the impact of residual fatigue on the present study parameters, morning and evening sessions have been randomly conducted on separate days as often conducted in the previous studies on the effect of time-of-day. In response to your concern about the two match days, we conducted a statistical analysis comparing initial values of all the parameters at both times of the day, and we did not find any significant difference. Comparison between baseline data being not the main outcome of this study, statistical results of baseline comparisons have not been included in the paper. Concerning the variability in participants' psychological conditions, we used the felt arousal scale [12] to evaluate the perceived arousal of the participants, and we did not find any significant change.

We have added in the section materials the description of the test:

“Felt Arousal Scale (FAS):

A single item measuring perceived arousal, the FAS [52] ranges from 1 (low arousal) to 6 (high arousal). Arousal is a physiological and mental condition that can be expressed in a variety of ways. High arousal is characterized by excitement, nervousness, or anger, and low arousal is characterized by calmness, relaxation, or boredom [52].”

 We added also the results of FAS in the table 4:

3.4. psychological state

Statistics analyzes (Table5) showed no significant interaction (TOD × Match) for FAS.

Table 5. Variation of psychological state before and after WFM in two moments of the day (Morning VS Evening) (values are mean ± SD)

Means ± SD

Interaction Time of day × Match

Before

After

Δ

Z

P-value

Cohen’s d

FAS

Morning

3.32±0.95

2.89±0.74

0.42±0.84

0.104

0.91

0.00

Evening

3.37±0.96

2.95±1.22

0.42±1.02

FAS: Felt arousal scale Δ: difference between after and before

The following sentence has been added to the discussion: “Moreover, the participant's psychological condition did not vary significantly between morning and evening sessions, as assessed by the FAS. From the FAS results, it seems that the diurnal variation of some physiological parameters in the present study were not dependent on the psychological state of the participants”

  • Variations in physiological and physical measurements may depend on the intensity of the activity (usually, refereed in terms of traveled distance, number of sprints, and number of steps). Are the inter-subject results affected by the intensity of the activity? How do you ensure that variations in glycemia, perceived exertion, and HRR60s are not related to the higher/lower intensity of the activity? Detail this aspect in the manuscript.

Response: Thank you for your comments. Our study was designed to explore the impact of time of day on physiological and physical measurements during walking football, enabling participants to self-regulate the intensity. A similar recent study conducted by the same research group (Please see reference [13] in the bottom of this text) examined the diurnal variation of self-paced cycling performance in trained subjects. Our focus here was on the natural diurnal variability in recreational football physiological responses, rather than the fixed intensity of the activity. Some further explanations are added to the discussion section in the revised version of our manuscript:

“In our study, we did not find any difference in MET values (4.85±0.62 in the morning and 4.9±0.46 in the evening) during WFM, which corresponds to the moderate exercise intensity category according to the established MET classification, where activities are classified as light intensity (<3 METs), moderate (3-6 METs), or vigorous (>6 METs) [14, 15].  Additionally, we did not find any significant change in Mean HR at both times of day (133.54 beat/min in the morning and 136.54 in the evening). The fact that the morning and evening values are similar suggests that there hasn't been any change in the level of activity throughout the day. Which indicates that the variations in blood glucose, and HRR60 are not related to the variability of exercise intensity between both sessions.”

1.) Abstract:

The abstract includes several abbreviations, which are difficult to read and consequently to interpret the results. Please, reduce the use of abbreviations throughout the manuscript.

Response: We limited the abstract to only two abbreviations to improve readability and facilitate clearer interpretation of the results.

  1. Introduction:
  • 1 Line 36: It states two times the term “associated”. Please, rephrase the sentence.

Response: The sentence has been rephrased as suggested; here is the new form with changes highlighted:

“Obesity often associated with sedentary behavior [1, 2], is linked to a high risk of diabetes mellitus, cardiovascular and pulmonary diseases, and cancers”.

  • 2 Line 52: Indicate the meaning and/or the differences of walking football when compared to regular walking. Refer to the speed and cadence differences.

Response: 

  • Thank you for your comments. We think that we cannot directly compare the cadence of WF with regular walking because they are different types of exercises. WF involves intermittent activity with frequent direction changes and walking pace (with an acyclic pattern), while regular walking is continuous and steady (with a cyclic pattern). Furthermore, cadence is influenced by various factors such as age, fitness level, and overall health. It's noteworthy that some studies have explored the chronic benefits of recreational football in comparison to running. In response to your suggestion, we have incorporated a sentence in the introduction to elucidate the cadence of WF and the pattern of walking employed in our study:

“Players should not run during WFM and must always walk (i.e. have at least one foot in contact with the ground at all times) [16]. Research has demonstrated that during a 1-hour WFM, participants covered a distance of 2.5 miles [17], maintained a walking cadence averaging 44-63 steps per minute [18], and an average of 100 changes in direction [19].”

  • Concerning the cadence of regular walking we have already mentioned in the discussion the recommendation of the necessary cadence of walking.

  • 3 Lines 68-70 state the gap in the literature that this paper aims to address. However, the studies 32 and 33 are not from the last years. The reviewer is not confident that “the relation between recreational exercise (e,i, WF) at different TOD and health risks” has still not been covered in the literature. Detail on related studies and open-research challenges according to the manuscript’s goal. Line 70: Attention to the term “e,i,”. The authors are encouraged to perform a careful English revision.

Response:

We thank the Reviewer for this comment. We would kindly refer the reviewer to our responses to his/her first comment where we detailed our search strategy. Moreover, we have added a recent study from 2023 to the studies 32 and 33.

  • We have correct the term “e,i,” and change it with “i.e.,”

  1. Methods:
  • 1) Provide a rationale for including only participants aged 30 years old. Complete the inclusion criteria regarding the gender, since the participants only include male subjects. Complete Table 1 with information about the exclusion criteria, i.e., indicate the number of participants excluded for each criterion.

Response:

  • The rationale for including only participants aged 30 years old: People go through significant changes in their lives during this phase, like starting families and careers. A more sedentary lifestyle and decreased physical activity may result from these changes, which may have a substantial effect on health outcomes [20, 21], and lead to many chronic health conditions, including metabolic disorders and cardiovascular disease [20, 21]. To avoid potential bias, we excluded elderly people and women from our study, to ensure the homogeneity of the population selected.
  • Inclusion criteria: we have added a line in Table 1:

Sex

Male

  • This part has been updated in figure 1:

  • 2 Line 86: replace “de-tailed”.

Response: “de-tailed” was correct and replaced with: detailed

  • 3) Detail the procedures of “10-minute passive recovery”. Were the participants free to self-select the activities during the resting period?

Response: We added this sentence to the description of the protocol in the section 2.2. Experimental design: “During the recovery period, participants are encouraged to engage in activities such as stretching and hydration.”

  • 4) Provide a rationale for the selected schedule of the times of day. Indicate the assessment time for pre-match and post-match.

Response:

  • According to previous studies, circadian fluctuations affect many physiological processes that can affect how the body reacts to acute exercise, including hormonal secretion and cardiovascular reactivity [22-25]. In this context, it has been shown that the unfavorable timing of physical activity identified closely aligns with the morning around 06h am to 12h pm and evening (06h pm –10h pm) peaks for cardiovascular risk [11, 26, 27]. Previous studies showed that there is a time-of-day specific cardiovascular reactivity to exercise, with the largest vagal withdrawal occurring at approximately 09:00 [28], catecholamine reactivity peaks occurring at approximately 09h am and 09h pm [28], and systolic blood pressure recovering more quickly from exercise in the late afternoon around 05h pm compared to the early morning (08h30 am)[29]. Evaluating WFM effects on these times of day allowed us, among others, to evaluate the safety of this activity across time of day mainly for cardiovascular risk parameters (i.e., HRV, HRR60). Additionally, previous studies have also shown that physical performance varies depending on the time of day, where it was higher in the late afternoon-early evening, coinciding with the peak of core body temperature between 04h pm and 06h pm [30]. Thus, our goal was to investigate the impact of daily fluctuations of physiological response and add to the body of knowledge regarding the acute response to physical activity, specifically the walking football.
  • The assessment time has been added in the new version: 2. Experimental design

“Participants are required to arrive 30 minutes before the beginning of each session. The morning session consisted of pre-match assessments from 06h30 to 07h am and post-match assessments from 08h to 08h30 am. The evening session followed the same pattern, with pre-match assessments from 05h00 to 05h30 pm and post-match assessments from 06h30 to 07:00 pm.”

3.5 Measurements:

  • indicate the sampling frequency and if they were synchronized.

Response:

We appreciate your feedback. We have specified the sampling frequency and the synchronization method for the data collection (HR, HRV and match parameters using HR sensors and ActiGraph GT3X) as detailed below:  

“R-R-intervals have been recorded using a modified protocol of Kammoun et al [26], with a sampling rate of 1000 Hz.”

“The accelerometers' initial data were collected at a sampling frequency of 30 Hz.”

  • Indicate the sensor system used to measure the “modified agility t-test”.

Response: We have added: “Using a stopwatch (C500B) to precisely measure the time taken during the test.”

  • What is nbs in line 173?

Response: nbs it means “number of steps”. It was added in line 202 in the revised version.

  • How was energy expenditure computed from accelerometer data?

Response:

The ActiGraph GT3X (ActiGraph Pensacola, FL, USA, www.actigraphcorp.com) is the pinnacle of scientific accelerometers (ActiGraph LLC, Pensacola, FL, USA). The sensitive triaxial accelerometers can store high-resolution, raw, unfiltered acceleration impulses for a considerable amount of time [31]. The ActiGraph monitor has been widely utilized for several purposes, including validation for the evaluation of physical activity (Energy Expenditure, step counts) [32, 33] and in comparison, with other activity trackers [34, 35]. In some research, ActiGraph monitors are also considered the gold standard for evaluating physical activity [36, 37]. In this context, a recent systematic review indicates that there are 235 published articles assessing sedentary time, physical activity, energy expenditure, or sleep using the ActiGraph GT3X [38]. ActiLife software has 12 different MET algorithms that can be used to calculate the average hourly and daily metabolic rate for a given dataset [39]. Counts per minute, a measurement of the number of movements the ActiGraph GT3X records, are used to compute the MET rate [39]. The individual's body mass is then multiplied by the MET rate to determine the energy expenditure in kilocalories per hour [39]. All details are mentioned on this site: https://url-r.fr/dXAuy

  • The authors should revise section 2.3 to detail the meaning of all measurements, the equipment, and the procedure used for their acquisition.

Response: Thank you very much for this comment. Section 2.3 was rewritten as suggested

  • 6) Indicate the level of significance.

Response:  This sentence has been added in the section “2.5. Statistical Analysis”: 

Statistical significance was considered when p <0.05.

  • 7) In line 190, the authors refer to a controlled parallel trial. However, based on the text reported in section 2.2., it seems that all participants performed the four matches (2 time-of-day periods for 2 days). Clarify this aspect in the manuscript.

Response:  We acknowledge the confusion regarding the reference to a controlled parallel trial. We would like to clarify that our study design was not a controlled parallel trial but rather a counter-balanced trial where the measurements were conducted before (control) and after (experimental) WF match in morning and evening sessions. The two sessions were separated by a 7-day recovery period, one in the morning and the second in the evening (to compare between both sessions). We have revised the wording in section 2.2 to accurately reflect this design.

The following sentence has been added: “The study was designed as a counter-balanced trial, where each participant was exposed to both experimental conditions (morning exercise compared to evening exercises).” The following sentence has been removed: “The study design was a controlled parallel trial, with realization of all the experimental conditions in the same period to minimize the effects of external conditions.”

  1. Results
  • 1) Revise the section to avoid misreading. In Line 220, verify the number of table. In line 224, the term “a significant” is reported without reference to a measure. In Table 3, the footnote symbols are not indicated within the results (lines).

Response:

  • The number of table has changed from “table 2” to “table 3”.
  • We have revised the manuscript, and the term “a significant” has been removed. This was an oversight on our part during the editing process.
  • In table 3 we remove: “a: significantly different from before”.

  • 2) The reviewer recommends the authors to highlight the significant values in Tables 3 and 4. It would facilitate the results’ interpretation.

Response: significant values in tables 3 and 4 has been changed on mode “Bold”.

  1. Discussion
  • 1) Use of the term “Interaction” instead of “relationship”

Response: We have changed the term “relationship” with “interaction” as suggested.

  • 2) Line 352: replace “the present” by “The present”.

Response: We have changed the term “the present” with “The present”.

  • 3) Remove the term “However” in line 390. It does not seem related to the previous sentence.

Response: We have removed the term “however” and replaced it with “in contrast”.

  • The authors stated that “While the heart rate recovery after 1 minute (HRR60s) of the match was lower in 28 the evening from the morning (p=0.048). In conclusion, the present study demonstrated the safety 29 of practicing WFM at different TOD.” Please, detail how reached such a conclusion from the achieved results.

Response:

We appreciate your feedback and recognize the need for clarity in our conclusion.

We have clarified the conclusion in the abstract of the revised version:

“Overall, walking football practice seems to be safe whatever the time of day. Furthermore, HRR60, glycemia and mean blood pressure values were lower in the evening compared to morning, suggesting that evening exercise practice could be safer in individuals with higher weight.”

We have added this explanation in the discussion to more explicate how it was safe based on HRR60s:

 “Some studies have established a threshold of 12 beats per minute after exercise to define a low HRR value [40], which is the difference between peak exercise HR and HRR60 [41]. In this context, in a previous study, HRR60s was found to be similar after maximal exercise in the morning (27 ± 7 bpm) and evening (29 ± 7 bpm) among hypertensive men [42]. Notably, a correlation exists between mortality risk and low HRR values [43]. All these findings contribute to the understanding of the safety of exercise based on HRR values.”

  • Comments on the Quality of English Language
  • Please, revise the manuscript. Some aspects were mentioned in the previous comment section. Commas are missing in various sentences.

Response: We have checked all the manuscript and made the necessary modification.  We hope that the quality of English is better in the revised version

  1. Crotty, N.M., et al., Reliability and Validity of the Lactate Pro 2 Analyzer. Measurement in Physical Education and Exercise Science, 2021. 25(3): p. 202-211.
  2. Pariente Rodrigo, E., et al., [Accuracy and reliability between glucose meters: A study under normal clinical practice conditions]. Semergen, 2017. 43(1): p. 20-27.
  3. Sassi, R.H., et al., Relative and absolute reliability of a modified agility T-test and its relationship with vertical jump and straight sprint. J Strength Cond Res, 2009. 23(6): p. 1644-51.
  4. Chow, G.C.-C., Y.-H. Kong, and W.-Y. Pun, The Concurrent Validity and Test-Retest Reliability of Possible Remote Assessments for Measuring Countermovement Jump: My Jump 2, HomeCourt & Takei Vertical Jump Meter. Applied Sciences, 2023. 13(4): p. 2142.
  5. Shahidi, S.H., et al., Validity and Reliability of Isometric Muscle Strength using the Powrlink Portable Device. International Journal of Strength and Conditioning, 2023. 3(1).
  6. Santos-Lozano, A., et al., Intermonitor variability of GT3X accelerometer. Int J Sports Med, 2012. 33(12): p. 994-9.
  7. Ravier, G., Quantifying internal workload during training drills in handball players: comparison between heart rate and perceived exertion based methods. Movement & Sport Sciences - Science & Motricité, 2022(118): p. 15-22.
  8. Weitzer, J., et al., Effect of time of day of recreational and household physical activity on prostate and breast cancer risk (MCC-Spain study). International Journal of Cancer, 2021. 148(6): p. 1360-1371.
  9. Shen, B., et al., Effects of exercise on circadian rhythms in humans. Frontiers in Pharmacology, 2023. 14: p. 1282357.
  10. Janssen, I., et al., Timing of physical activity within the 24-hour day and its influence on health: a systematic review. Health Promotion and Chronic Disease Prevention in Canada: Research, Policy and Practice, 2022. 42(4): p. 129-138.
  11. Feng, H., et al., Associations of timing of physical activity with all-cause and cause-specific mortality in a prospective cohort study. Nat Commun, 2023. 14(1): p. 930.
  12. Svebak, S. and S. Murgatroyd, Metamotivational dominance: A multimethod validation of reversal theory constructs. Journal of Personality and Social Psychology, 1985. 48(1): p. 107-116.
  13. Souissi, W., et al., Higher evening metabolic responses contribute to diurnal variation of self-paced cycling performance. Biology of Sport, 2022. 39(1): p. 3-9.
  14. Ainsworth, B.E., et al., 2011 Compendium of Physical Activities: a second update of codes and MET values. Med Sci Sports Exerc, 2011. 43(8): p. 1575-81.
  15. Ainsworth, B.E., et al., Compendium of physical activities: an update of activity codes and MET intensities. Med Sci Sports Exerc, 2000. 32(9 Suppl): p. S498-504.
  16. FA, T., WALKING FOOTBALL LAWS

OF THE GAME MADE SIMPLE.

  1. Ferguson, R., Walking Soccer / Football Has Arrived | 50+ World - 50+ World. 2019.
  2. Salle, D.D.A., R.U. Newton, and D.P. Heil, The Walking Cadence Threshold Associated With A Moderate Metabolic Intensity During Competitive Walking Football: 2554. Medicine & Science in Sports & Exercise, 2023. 55(9S): p. 844-844.
  3. Harper, L.D., et al., The Physiological, Physical, and Biomechanical Demands of Walking Football: Implications for Exercise Prescription and Future Research in Older Adults. J Aging Phys Act, 2019: p. 1-11.
  4. Lavie, C.J., et al., Sedentary Behavior, Exercise, and Cardiovascular Health. Circ Res, 2019. 124(5): p. 799-815.
  5. Winkler, S., A. Hebestreit, and W. Ahrens, [Physical activity and obesity]. Bundesgesundheitsblatt Gesundheitsforschung Gesundheitsschutz, 2012. 55(1): p. 24-34.
  6. Ayyar, V.S. and S. Sukumaran, Circadian rhythms: influence on physiology, pharmacology, and therapeutic interventions. Journal of Pharmacokinetics and Pharmacodynamics, 2021. 48(3): p. 321-338.
  7. Aoyama, S. and S. Shibata, Time-of-Day-Dependent Physiological Responses to Meal and Exercise. Frontiers in Nutrition, 2020. 7: p. 18.
  8. Skene, D.J. and J. Arendt, Human circadian rhythms: physiological and therapeutic relevance of light and melatonin. Annals of Clinical Biochemistry: International Journal of Laboratory Medicine, 2006. 43(5): p. 344-353.
  9. Maywood, E.S., et al., Minireview: The Circadian Clockwork of the Suprachiasmatic Nuclei—Analysis of a Cellular Oscillator that Drives Endocrine Rhythms. Endocrinology, 2007. 148(12): p. 5624-5634.
  10. Muller, J.E., et al., Circadian Variation in the Frequency of Onset of Acute Myocardial Infarction. New England Journal of Medicine, 1985. 313(21): p. 1315-1322.
  11. Casetta, I., et al., Patient Demographic and Clinical Features and Circadian Variation in Onset of Ischemic Stroke. Archives of Neurology, 2002. 59(1): p. 48.
  12. Scheer, F.A.J.L., et al., Impact of the human circadian system, exercise, and their interaction on cardiovascular function. Proceedings of the National Academy of Sciences of the United States of America, 2010. 107(47): p. 20541-20546.
  13. Qian, J., et al., The circadian system modulates the rate of recovery of systolic blood pressure after exercise in humans. Sleep, 2020. 43(4).
  14. Kline, C.E., et al., Circadian variation in swim performance. Journal of Applied Physiology, 2007. 102(2): p. 641-649.
  15. Guediri, A., et al., Comparison of Energy Expenditure Assessed Using Wrist- and Hip-Worn ActiGraph GT3X in Free-Living Conditions in Young and Older Adults. Front Med (Lausanne), 2021. 8: p. 696968.
  16. Reid, R.E.R., et al., Validity and reliability of Fitbit activity monitors compared to ActiGraph GT3X+ with female adults in a free-living environment. J Sci Med Sport, 2017. 20(6): p. 578-582.
  17. Barwais, F.A., et al., ActiGraph GT3X determined variations in “free-living” standing, lying, and sitting duration among sedentary adults. Journal of Sport and Health Science, 2013. 2(4): p. 249-256.
  18. Tudor-Locke, C., T.V. Barreira, and J.M. Schuna, Jr., Comparison of step outputs for waist and wrist accelerometer attachment sites. Med Sci Sports Exerc, 2015. 47(4): p. 839-42.
  19. Webber, S.C. and P.D. St John, Comparison of ActiGraph GT3X+ and StepWatch Step Count Accuracy in Geriatric Rehabilitation Patients. J Aging Phys Act, 2016. 24(3): p. 451-8.
  20. Moon, Y., et al., Monitoring gait in multiple sclerosis with novel wearable motion sensors. PLoS One, 2017. 12(2): p. e0171346.
  21. Chandrasekar, A., et al., Preliminary concurrent validity of the Fitbit-Zip and ActiGraph activity monitors for measuring steps in people with polymyalgia rheumatica. Gait Posture, 2018. 61: p. 339-345.
  22. Migueles, J.H., et al., Accelerometer Data Collection and Processing Criteria to Assess Physical Activity and Other Outcomes: A Systematic Review and Practical Considerations. Sports Med, 2017. 47(9): p. 1821-1845.
  23. ActiGraph Support Center.
  24. Cole, C.R., et al., Heart-rate recovery immediately after exercise as a predictor of mortality. N Engl J Med, 1999. 341(18): p. 1351-7.
  25. Barak, O.F., et al., Heart rate recovery after submaximal exercise in four different recovery protocols in male athletes and non-athletes. J Sports Sci Med, 2011. 10(2): p. 369-75.
  26. Brito, L., et al., Time of day affects heart rate recovery and variability after maximal exercise in pre-hypertensive men. Chronobiol Int, 2015. 32(10): p. 1385-90.
  27. Cole, C.R., et al., Heart-Rate Recovery Immediately after Exercise as a Predictor of Mortality. New England Journal of Medicine, 1999. 341(18): p. 1351-1357.

Round 2

Reviewer 2 Report

Comments and Suggestions for Authors

The authors have discussed all my previous comments. Some of them were properly addressed in the revised manuscript. There is still room for improving the introduction, namely the statement in lines 76 to 78, considering the author’s answer to my first comment.  

Author Response

Dear Reviewer,

Thank you very much for your comments.

The corrections have been highlighted in red.  The introduction has been corrected as suggested. Some little imperfections and typos have been corrected in the text.

Best regards,
